# Caries risk assessment using different Cariogram models. A comparative study about concordance in different populations— Adults and children

**Maria Grazia Cagetti**[1], **Giuliana Bontà**[1‡], **Juan Sebastian Lara**[2‡], **Guglielmo Campus** [ID][3,4,5] *

**1** Department of Biomedical, Surgical and Dental Sciences, University of Milan, Milan, Italy, **2** Department of Cariology, Operative Dentistry and Dental Public Health, Indiana University School of Dentistry, Indianapolis, IN, United States of America, **3** Department of Restorative, Preventive and Pediatric Dentistry, University of Bern, Bern, Switzerland, **4** Department of Surgery, Microsurgery and Medicine Sciences, School of Dentistry, University of Sassari, Sassari, Italy, **5** Department of Pediatric, Preventive Dentistry and Orthodontics, School of Dentistry, Sechenov University, Moscow, Russia

☯ These authors contributed equally to this work.
‡ GB and JSL also contributed equally to this work.
* guglielmo.campus@zmk.unibe.ch

**Data Availability Statement:** All relevant data are within the manuscript and its Supporting Information files.

**Funding:** No funding was obtained for this study.

## Abstract

This methodological survey aimed to verify whether there is concordance among several Cariogram different risk models at different thresholds, comparing both children and adult populations and how each risk/protective factor weight on the overall caries risk profile. Three groups' data (two in children and one in adults) were obtained from previous studies, while a fourth, in young adults, was *ad hoc* enrolled. Different caries risk levels were assessed: a) three risk categories with two different thresholds as: "low risk" = 61–100% or 81–100% chance to avoid caries, "moderate risk" = 41–60% or 21–80% and "high risk" = 0–40% or 0–20%, named model 1 and 2; b) four risk categories with two different thresholds as: "low risk" = 61–100% or 76–100%, "moderate/low risk" = 41–60% or 51–75%; "moderate/high risk" = 21–40% or 26–50% and "high risk" = 0–20% or 0–25%, model 3 and 4; c) five risk categories as: "very low risk" = 81–100%; "low risk" = 61–80% "moderate risk" = 41–60%; "high risk" = 21–40% and "very high risk" = 0–20%, model 5. Concordance of the different Cariogram risk categories among the four groups was calculated using Cohen's kappa. The weight of the association between all Cariogram models toward the Cariogram risk variables was evaluated by ordinal logistic regression models. Considering Cariogram model 1 and 2, Cohen's Kappa values ranged from 0.40 (SE = 0.07) for the young adult group to 0.71 (SE = 0.05) for the adult one. Cohen's Kappa values ranged from 0.14 (SE = 0.03 p<0.01) for the adult group to 0.62 (SE = 0.02) for the two groups of children in models 3 and 4. Statistically significant associations were found for all Cariogram risk variables excepting Fluoride program in models 4 and 5 and the overall risk on children's samples. Caries experience showed a quite variable weight in the different models in both adult groups. In the regression analyses, adult groups' convergence was not always achievable since variations in associations between caries risk and different risk variables were

**Competing interests:** The Authors certify that there is no conflict of interest with any financial organization regarding the material discussed in the manuscript.

**Abbreviations:** dmft, Decayed, missed, filled teeth (primary dentition); DMFT, Decayed, missed, filled teeth (permanent dentition); OR, Odds ratio; 95% CI, 95% Confidence interval.

narrower compared to other samples. Significant differences in caries risk stratification using different thresholds stands out from data analysis; consequently, risk assessments need to be carefully considered due to the risk of misleadingly choosing preventive and research actions.

## Introduction

Assessing and classifying individual caries risk is an essential step in the caries diagnosis and decision-making process. Caries risk assessment allows to achieve the best non-invasive and operative options, as well as establishing recall intervals [1, 2].

Caries Risk Assessment (CRA) should be an integral part of the treatment regimens, in order to plan an individualized caries prevention strategy. However, routinely the patient caries risk profile is not always determined and, when performed, the risk level is not always linked to the adopted preventive and intervention strategies [3, 4].

From the 2000s onwards, a variety of standardized tools were developed and tested in different populations [5, 6]. All multifactorial risk tools include a different combination of risk and protective factors, attributing to each variable a different weight in the caries risk calculation [7]. Despite this, the scientific evidence on CRA tools validity remains limited [5, 6, 8]. Overall, the Cariogram is the only multidimensional tool verified in children and adults in few good quality studies, even if its accuracy level was found to be limited in different age groups [5, 9]. Moreover, the Cariogram has been applied in different ways: considering seven to ten risk variables, considering three to five risk categories and finally, considering different thresholds to define each risk category [10–12]. These thresholds can lead to a loss of information because the entire range of predicted probabilities is not used, but also to substantial bias because the thresholds can occur as data-driven instead of pre-specified clinical data [9]. In particular, for risk assessment, an inappropriate use of a diagnostic tool can theoretically lead to both clinical and research bias. In fact, an underestimation or overestimation of the risk can lead to the choice of an inadequate preventive/therapeutic regime and/or to an incorrect enrolment of a subject in a study sample.

While the effectiveness of the Cariogram used with a reduced number of factors, mostly excluding salivary factors [13–17], has been validated, while the use of different risk categories considering different ranges of percentages to avoid caries was not investigated. In view of the foregoing, the present methodological study aimed, primary, to verify if there is concordance among the Cariogram caries risk categories at different thresholds in children and adults; and secondly, how much each risk/protective variable weight for the different overall caries risk profile in different age groups.

## Material and methods

### Samples

Four sample groups were considered in the comparisons. Two were obtained from previous studies in children (available at https://www.karger.com/Article/FullText/334932 and https://www.tandfonline.com/doi/full/10.1080/00016350902740498, respectively) [13, 16], one in adults (available at https://www.scielo.br/j/bor/a/ZqcrsmPRxNJQsYPV5nnNTTS/?lang=en) [17], and the fourth one, which included young adults, was *ad hoc* enrolled. Unlike the two groups of adults who are two distinct samples in which the risk assessment was carried out one

time, the two samples of children are the same group of children whose caries risk was evaluated and re-evaluated after two years. Regarding the published papers [13, 16, 17], the abstracts can be easily retrieved on Pubmed.

Ethical issues were addressed as follows: the retrospective data analysis of archived samples the IRB of the University of Sassari waived the approval (IRB Sassari 20/04/2020); regarding the *ad hoc* sample the study proposal was accepted by the Ethical Committee of the University Hospital Milan (Comitato Etico Milano 1 n. 2019/ST/167).

All the surveys from which the data derives, were conducted by the same research group; in particular, two authors of the present survey acted as examiners/benchmarks in all surveys: MGC ($A_{young}$ and $A_{old}$ surveys) and GC ($C_{young}$ and $C_{old}$ surveys as examiner and $A_{young}$ and $A_{old}$ surveys as benchmark).

In children's groups, a reduced Cariogram model (excluding saliva parameters: buffer capacity and saliva secretion rate) was build-up to assess the forecast probabilities of new caries. Two years later, the same sample was re-examined and the caries risk re-calculated. The children samples consisted of 957 subjects aged 7–9 years ($C_{young}$) and 862 children aged 9–11 years ($C_{old}$).

The adult sample ($A_{old}$) included 480 subjects aged 30–45 years. Caries risk was calculated through the full Cariogram model. For the last sample of young adults ($A_{young}$), undergraduate second and third-year dental students, School of Dentistry, University of Milan, Italy, were invited to participate. This convenience young adult sample consisted of 86 subjects (mean age 23.37 ± 3.11 years).

## Data collection

Data for samples $C_{young}$, $C_{old}$ and $A_{old}$, were extracted from the previously published studies. Data collection for sample $A_{young}$ followed the same procedures in the adult population study [17], which included: a standardized high-structured self-administered questionnaire (in Italian and English language S1 and S2 Appendices) [18] collecting data on age, oral hygiene habits (frequency of toothbrushing), dietary habits (daily intake of sugary foods and drinks), fluoride exposure in addition to toothpaste and other remineralizing and antibacterial compounds, frequency of dental check-ups, presence of systemic diseases, and use of medications. Additionally, participants received clinical examinations by two examiners (GB and MGC) to determine caries prevalence at dentinal level, caries experience (Decayed, Missing and Filled Teeth index) [19] and plaque index [14]. The examiners were *ad hoc* calibrated on 15 subjects reporting a strong Cohen's Kappa scores both for inter- and intra-examiner reliability (0.87/ 0.89 for inter-examiner and 0.85 for intra-examiner reliability) [20–22]. Finally, saliva chairside tests were performed following the manufacturer's instructions to evaluate the salivary oral streptococci and *Lactobacillus spp*. counts (CRT bacteria, Ivoclar Vivadent AG, Schaan, Liechtenstein), and buffer capacity (CRT Buffer, Ivoclar Vivadent AG, Schaan, Liechtenstein). The raw-data file was uploaded as S1 Data.

## Risk assessment

Caries risk profiles were calculated using the Cariogram for all subjects in the different groups. Cariogram risk for samples $C_{young}$ and $C_{old}$ was calculated including seven risk/protective variables: caries experience, related diseases, diet content, diet frequency, plaque amount, oral streptococci count and fluoride program. For samples $A_{young}$ and $A_{old}$, in addition to the previous factors, saliva secretion rate and buffer capacity were also included.

The caries risk level was assessed with five different models (Table 1): models 1 and 2—following three risk categories with two different thresholds (% chance of avoid caries) as: "low

**Table 1. Cariogram models and cutt-off values considered.**

| Cariogram | Risk levels (n) | Cut-offs (% of chances to avoid caries) and risk categories | | | | |
|---|---|---|---|---|---|---|
| | | Low | Moderate | High | | |
| Model 1[a] | 3 | 61–100% | 41–60% | 0–40% | | |
| Model 2[b] | 3 | 81–100% | 21–80% | 0–20% | | |
| | | Low | Low/Moderate | Moderate/High | High | |
| Model 3[c] | 4 | 61–100% | 41–60% | 21–40% | 0–20% | |
| Model 4[d] | 4 | 76–100% | 51–75% | 26–50% | 0–25% | |
| | | Very low | Low | Moderate | High | Very high |
| Model 5[e] | 5 | 81–100% | 61–80% | 41–60% | 21–40% | 0–20% |

[a]Petsi G, Gizani S, Twetman S, Kavvadia K. Cariogram caries risk profiles in adolescent orthodontic patients with and without some salivary variables. Angle Orthod. 2014 Sep;84(5):891–5; Al Mulla AH, Kharsa SA, Kjellberg H, Birkhed D. Caries risk profiles in orthodontic patients at follow-up using Cariogram. Angle Orthod. 2009 Mar;79(2):323–30.

[b]Zukanovic A. Caries risk assessment models in caries prediction. Acta Med Acad. 2013;42:198–208.

[c]Chang J, Kim HY. Does caries risk assessment predict the incidence of caries for special needs patients requiring general anesthesia? Acta Odontol Scand. 2014;72:721–8; Petersson GH, Fure S, Bratthall D. Evaluation of a computer-based caries risk assessment program in an elderly group of individuals. Acta Odontol Scand. 2003;61:164–71.

[d]Twetman S, Petersson GH, Bratthall D. Caries risk assessment as a predictor of metabolic control in young Type 1 diabetics. Diabet Med. 2005;22:312–5.

[e]Campus G, Cagetti MG, Sale S, Carta G, Lingström P. Cariogram validity in schoolchildren: a two-year follow-up study. Caries Res. 2012;46:16–22; Petersson GH, Twetman S. Caries risk assessment in young adults: a 3 year validation of the Cariogram model. BMC oral health. 2015;15:17.

risk" = 61–100% or 81–100%, "moderate risk" = 41–60% or 21–80% and "high risk" = 0–40% or 0–20%, respectively; models 3 and 4—following four risk categories with two different thresholds: "low risk" = 61–100% or 76–100%, "moderate/low risk" = 41–60% or 51–75%; "moderate/high risk" = 21–40% or 26–50% and "high risk" = 0–20% or 0–25%, respectively; and finally, model 5—in five categories as follow: "very low risk" = 81–100%; "low risk" = 61–80% "moderate risk" = 41–60%; "high risk" = 21–40% and "very high risk" = 0–20% chance to avoid caries, respectively.

## Statistical analysis

Cohen´s Kappa values were calculated to assess the agreement of the Cariogram risk categories among the 4 groups. Kappa statistics were tested using z test at a significance level of $\alpha = 0.01$. Ordinal logistic regression models were built to evaluate the weight of the associations, via Odds Ratio and $_{95\%}$Confidence Intervals, of the variables inserted on each Cariogram model toward Cariogram categorizations.

## Results

Table 2 shows the demographic characteristics and caries figures of all groups. The mean caries experience (DMFT/dmft) ranged from 2.37±4.05 in the $C_{young}$ (including both primary and permanent dentitions) to 3.25±4.15 in the $A_{old}$, with a D/d component ranging from 1.38 ±2.62 to 0.24±0.72, respectively.

Comparisons among groups and the different risk categories and thresholds are displayed in Table 3. Caries risk profiles were statistically significant different for all age groups (p values ranging from <0.01 to 0.02) when considering three (low, moderate and high) and four (low, moderate/low, moderate/high and high) risk categories. Particularly, Cohen´s Kappa values ranged from 0.71 (SE = 0.05) to 0.40 (SE = 0.07) in the $A_{old}$ and the $A_{young}$ groups, respectively. Children groups showed intermediate values. On the other hand, when four risk categories

**Table 2. Description of the sample.**

| Samples | $C_{young}$ | $C_{old}$ | $A_{young}$ | $A_{old}$ |
|---|---|---|---|---|
| | n = 957 | n = 862 | n = 86 | n = 480 |
| Gender | Males n = 473 (49.37) | Males n = 421 (48.84%) | Males n = 39 (45.35%) | Males n = 254 (52.92%) |
| | Females n = 485 (50.63) | Females n = 441 (51.16%) | Females n = 47 (54.65%) | Females n = 226 (47.08%) |
| Age | 6.49±0.71 years (range 7–9) | 8.16±0.94 years (range 9–11) | 23.37±3.11 years (range 21–27) | 40.72±9.41 years (range 30–45) |
| Caries-free | 409 (47.91%) | 368 (42.69%) | 26 (30.23%) | 175 (36.46%) |
| Caries Experience | 2.37±4.05 (range 0–19) | 3.22±4.49 (range 0–23) | 2.91±3.30 (0–13) | 3.25±4.15 (0–21) |
| D/d | 1.38±2.62 (range 0–17) | 2.51±3.78 (range 0–22) | 0.26±0.64 (range 0–3) | 0.24±0.72 (0–8) |
| M/m | — | — | 0.09±0.62 (range 0–5) | 0.49±1.54 (0–9) |
| F/f | 0.4±1.74 (range 0–7) | 0.63±2.21 (range 0–10) | 2.56±3.00 (range 0–11) | 2.51±4.56 (0–20) |

were considered, Cohen´s Kappa values ranged from 0.62 (SE = 0.02) to 0.14 (SE = 0.03 $p < 0.01$) in both children groups and in the $A_{old}$ group. Comparisons for the last risk model (including 5 categories: very low, low, moderate, high, very high) were not performed since only one set of thresholds was used.

**Table 3. Caries risk in the four samples using the five Cariogram models.**

| | Model 1 (3 Risk categories) | | | |
|---|---|---|---|---|
| | $C_{young}$ | $C_{old}$ | $A_{young}$ | $A_{old}$ |
| Low | 487 (50.94) | 449 (52.09) | 55 (63.95) | 12 (2.50) |
| Moderate | 195 (20.40) | 165 (19.14) | 20 (23.26) | 386 (80.42) |
| High | 274 (28.66) | 248 (28.77) | 32 (37.21) | 82 (17.08) |
| | **Model 2 (3 Risk categories)** | | | |
| | $C_{young}$ | $C_{old}$ | $A_{young}$ | $A_{old}$ |
| Low | 332 (34.73) | 309 (35.85) | 32 (37.21) | – |
| Moderate | 432 (45.19) | 378 (43.85) | 52 (60.47) | 479 (99.79) |
| High | 192 (20.08) | 175 (20.30) | 2 (2.33) | 1 (0.21) |
| *Cohen Kappa* | *K = 0.63 SE = 0.03 p<0.01* | *K = 0.63 SE = 0.02 p<0.01* | *K = 0.40 SE = 0.07 p<0.01* | *K = 0.71 SE = 0.05 p = 0.02* |
| | **Model 3 (4 Risk categories)** | | | |
| | $C_{young}$ | $C_{old}$ | $A_{young}$ | $A_{old}$ |
| Low | 487 (50.94) | 449 (52.09) | 55 (63.95) | 12 (2.50) |
| Low/Moderate | 195 (20.40) | 165 (19.14) | 20 (23.26) | 386 (80.42) |
| Moderate/High | 82 (8.58) | 73 (8.47) | 9 (10.47) | 81 (16.88) |
| High | 192 (20.08) | 175 (20.30) | 2 (2.33) | 1 (0.21) |
| | **Model 4 (4 Risk categories)** | | | |
| | $C_{young}$ | $C_{old}$ | $A_{young}$ | $A_{old}$ |
| Low | 332 (34.73) | 309 (35.85) | 37 (43.02) | – |
| Low/Moderate | 249 (26.05) | 213 (24.71) | 29 (33.72) | 152 (31.67) |
| Moderate/High | 183 (19.14) | 165 (19.14) | 15 (17.44) | 326 (67.92) |
| High | 192 (20.08) | 175 (20.30) | 5 (5.81) | 2 (0.42) |
| *Cohen Kappa* | *K = 0.62 SE = 0.02 p<0.01* | *K = 0.62 SE = 0.02 p<0.01* | *K = 0.44 SE = 0.07 p<0.01* | *K = 0.14 SE = 0.03 p<0.01* |
| | **Model 5 (5 Risk categories)** | | | |
| | $C_{young}$ | $C_{old}$ | $A_{young}$ | $A_{old}$ |
| Very Low | 332 (34.73) | 309 (35.85) | 55 (63.95) | – |
| Low | 155 (16.21) | 140 (16.24) | 20 (23.26) | 12 (2.50) |
| Moderate | 195 (20.40) | 165 (19.14) | –__ | 386 (80.42) |
| High | 82 (8.58) | 73 (8.47) | 9 (10.47) | 81 (16.88) |
| Very High | 192 (20.08) | 175 (20.30) | 2 (2.33) | 1 (0.21) |

A multinomial regression analysis was run, using caries risk categories as dependent variables, to evaluate the weight of the association (OR) between the overall risk measured with the different models and the risk/protective variables considered in each model (Table 4). In the $C_{young}$ and $C_{old}$ groups all considered risk variables were significantly associated to the caries risk profiles, except for fluoride program in models 4 and 5. However, each OR values ranged widely among models. While the weight of caries experience, oral streptococci and fluoride program remained quite stable in different models, diet content, diet frequency and plaque amount showed the widest variability in the weight of the associations. In $C_{young}$, diet content ranged from OR 2.48 $_{95\%}$CI 1.30 / 4.74 to OR 22.09 $_{95\%}$CI 6.23 / 78.34 in model 5 and 2, respectively; and plaque amount from OR 2.24 $_{95\%}$CI 1.53 / 3.28 to OR 299.53 $_{95\%}$CI 106.26 / 844.38 in model 1 and 2, respectively. Diet frequency for $C_{old}$ exhibited the highest value in model 1 (OR 16.06 $_{95\%}$CI 1.94 / 13.15) and the lowest in model 4 (OR 4.54 $_{95\%}$CI 1.90 / 10.86), while plaque amount showed the highest value in model 2 (OR 405.56 $_{95\%}$CI 143.68 / 1144.80) and the lowest in model 1 (OR 1.55 $_{95\%}$CI 1.01 / 2.36). In relation to $A_{young}$ group, caries experience showed a quite variable weight in the different models, reaching the highest value in model 3 (OR 32.49 $_{95\%}$CI 4.00 / 263.85), and the lowest in model 4 (OR 2.66 $_{95\%}$CI 1.56 / 4.52). All other risk variables resulted more strongly associated with risk profiles measured using model 1, with oral streptococci count and fluoride program showing the widest variations among models. Finally, given the low variance across risk variables and the reduced number of subjects in some risk categories, in the $A_{old}$ group for models 2 and 4, the regression analysis was only achieved for model 1, 3 and 5. In this group, variation in weight between caries risk and different risk variables were narrower less broad compared to the above described samples. Fluoride program reached the highest OR in model 1 (3.16 95%CI 1.45 / 6.91) and buffer capacity in model 3 and 5 (OR 7.28 95%CI 0.90 / 34.62, in both models).

## Discussion

This study was designed to evaluate the concordance among caries risk profiles evaluated with Cariogram at different risk categories and different thresholds. The study used previous published data (two samples of schoolchildren and one of adults) and original data from a fourth group of young adults. The significant difference in caries risk stratification using different thresholds stands out as the main outcome of the present study.

The Cariogram demonstrates the caries risk graphically and provides a percentage value illustrating the probability an individual has to avoid the development of lesions in the near future. Nevertheless, the weights attributed to the risk factors considered in the Cariogram did not appear to explain caries experience or caries activity in all groups [23].

When assessing the caries risk what is expected to forecast is the chance to develop or not the disease and, in this perspective, only the subsequent incidence of "true" caries can confirm or disprove the initial hypothesis. The Cariogram was initially tested in children considering five risk categories [10]. Afterwards, its validity was demonstrated in elderlies and adults using different Cariogram models [24, 25]. Lack of calibration is a common problem of the Cariogram software, as the predictive capacity developed using data from a certain population can decrease when the tool is applied to another population. In these conditions, a recalibration can improve predictive capacity [6].

From a public health perspective, when caries risk is assessed in population samples, it is useful to allocate each subject into a risk category, facilitating the implementation of large-scale preventive and management interventions. However, the need to allocate subjects into risk categories for community purposes, as emerges from the findings of the present methodological study, can lead to risk category allocation biases due to the number of risk categories

**Table 4. Multinomial regression analysis.** Cariogram risk categories as dependent variables.

| $C_{young}$ Group | | | | | |
|---|---|---|---|---|---|
| Risk variables | Model 1 OR (SE) $_{95\%}$CI | Model 2 OR (SE) $_{95\%}$CI | Model 3 OR (SE) $_{95\%}$CI | Model 4 OR (SE) $_{95\%}$CI | Model 5 OR (SE) $_{95\%}$CI |
| *Caries experience* | 1.51(0.06) 1.40 / 1.63 | 1.53 (0.07) 1.39 / 1.67 | 1.40 (0.05) 1.31 / 1.49 | 1.72 (0.07) 1.58 / 1.87 | 1.59 (0.06) 1.49 / 1.70 |
| *Related disease* | – | – | – | – | – |
| *Diet contents* | 7.49 (5.03) 2.01 / 27.93 | 22.09 (14.27) 6.23 / 78.34 | 7.55 (5.10) 2.01 / 28.36 | 9.27 (3.91) 4.05 / 21.18 | 2.48 (0.82) 1.30 / 4.74 |
| *Diet frequency* | 2.85 (0.38) 2.20 / 3.70 | 2.73 (0.46) 1.96 /3.80 | 3.17 (0.43) 2.43 / 4.13 | 2.99 (0.40) 2.29 / 3.87 | 2.67 (0.31) 2.12 / 3.36 |
| *Plaque amounts* | 2.24 (0.44) 1.53 / 3.28 | 299.53 (158.39) 106.26 / 844.38 | 6.13 (1.14) 4.26 / 8.83 | 127.92 (38.26) 71.17 / 229.90 | 11.71(2.00) 8.38 / 16.37 |
| *Mutans streptococci* | 2.32 (0.15) 2.05 / 2.62 | 2.71 (0.24) 2.28 / 3.22 | 2.32 (0.13) 2.07 / 2.61 | 3.02 (0.23) 2.60 / 3.50 | 2.43 (0.141) 2.18 / 2.73 |
| *Fluoride program* | 1.78 (0.09) 1.58 / 2.06 | 1.84 (0.08) 1.61 / 2.24 | 1.48 (0.04) 1.15 / 2.02 | 1.25 (0.05) 0.94 / 1.54 | 1.25 (0.05) 0.94 / 1.54 |

| $C_{old}$ Group | | | | | |
|---|---|---|---|---|---|
| Risk variables | Model 1 OR (SE) $_{95\%}$CI | Model 2 OR (SE) $_{95\%}$CI | Model 3 OR (SE) ($_{95\%}$CI) | Model 4 OR (SE) $_{95\%}$CI | Model 5 OR (SE) $_{95\%}$CI |
| *Caries experience* | 1.69 (0.09) 1.52 / 1.87 | 1.50 (0.08) 1.36 / 1.66 | 1.43 (0.05) 1.33 / 1.54 | 1.81 1.64 / 2.00 | 1.63 (0.06) 1.51 / 1.76 |
| *Related disease* | – | – | – | – | – |
| *Diet contents* | 3.25 (0.48) 2.43 / 4.35 | 3.27 (0.61) 2.27 / 4.70 | 3.84 (0.58) 2.85 / 5.17 | 3.23 (0.46) 2.43 / 4.28 | 3.20 (0.41) 2.49 / 4.12 |
| *Diet frequency* | 16.06 (17.33) 1.94 / 13.15 | 9.02 (6.00) 2.45 / 33.21 | 11.40 (12.23) 1.39 / 93.33 | 4.54 (2.02) 1.90 / 10.86 | 1.40 (0.49) 0.70 / 2.80 |
| *Plaque amounts* | 1.55 (0.33) 1.01 / 2.36 | 405.56 (214.73) 143.68 / 1144.80 | 5.51 (1.11) 3.71 / 8.17 | 127.19 (39.15) 69.58 / 232.50 | 12.27 (2.24) 8.57 / 17.56 |
| *Mutans streptococci* | 2.42 (0.17) 2.11 / 2.79 | 2.60 (0.24) 2.18 / 3.12 | 2.27 (0.14) 2.0 / 2.56 | 3.00 (0.24) 2.57 / 3.50 | 2.39 (0.14) 2.12 / 2.67 |
| *Fluoride program* | 2.03 (0.12) 1.71 / 2.82 | 1.43 (0.03) 0.95 / 1.95 | 1.73 (0.04) 0.88 / 2.70 | 2.01 (0.05) 1.09 / 3.10 | 2.03 (0.05) 1.11 / 3.26 |

| $A_{young}$ Group | | | | | |
|---|---|---|---|---|---|
| Risk variables | Model 1 OR (SE) $_{95\%}$CI | Model 2 OR (SE) P-value $_{95\%}$CI | Model 3 OR (SE) P-value ($_{95\%}$CI) | Model 4 OR (SE) $_{95\%}$CI | Model 5 OR (SE) $_{95\%}$CI |
| *Caries experience* | 17.60 (20.74) 3.17 / 34.66 | 5.83 (2.85) 2.23 / 15.18 | 32.49 (34.72) 4.00 / 263.85 | 2.66 (0.719) 1.56 / 4.52 | 10.45 (4.68) 4.35 / 25.13 |
| *Related disease* | – | – | – | – | – |
| *Diet contents* | 38.66 (55.28) 2.35 / 637.47 | 4.36 (2.33) 0.53 / 12.45 | 16.911 (16.30) 2.69 / 111.90 | 5.83 (2.26) 2.73 / 12.45 | 8.24 (3.86) 3.29 / 20.64 |
| *Diet frequency* | 35.13 (45.94) 2.71 / 455.92 | 3.66 (1.72) 1.45 / 9.20 | 8.44 (5.65) 2.27 / 31.34 | 3.32 (1.08) 1.76 / 6.27 | 5.18 (1.96) 2.46 / 10.88 |
| *Plaque amounts* | 55.69 (106.22) 1.32 / 339.80 | 1.81 (1.77) 0.26 / 12.37 | 22.90 (34.84) 1.162 / 451.47 | 2.70 (2.02) 0.63 / 11.68 | 3.35 (2.56) 0.74 / 15.04 |
| *Mutans streptococci* | 238.38 (429.67) 6.97 / 815.87 | 4.57 (2.30) 1.71 / 12.25 | 81.96 (100.84) 7.35 / 913.92 | 4.47 (1.70) 2.12 / 9.44 | 10.38 (5.17) 3.91 / 27.56 |
| *Fluoride program* | 126.60 (41.07) 11.07 / 3388.01 | 23.45 (21.48) 3.89 / 141.14 | 97.64 (182.57) 16.56 / 553.66 | 11.15 (5.77) 4.04 / 30.76 | 85.99 (71.36) 16.90 / 437.35 |
| *Saliva secretion* | 75.71 (22.75) 3.15 / 181.69 | 4.17 (0.91) 1.01 / 16.40 | 11.78 (1.37) 2.10 / 26.13 | 3.12 (1.40) 1.30 / 7.52 | 7.39 (3.85) 2.67 / 20.44 |
| *Buffer capacity* | 36.08 (60.46) 1.35 / 96.29 | 3.94 (1.60) 0.73 / 21.03 | 5.55 (1.29) 0.87 / 35.77 | 5.84 (2.76) 1.67 / 20.46 | 8.07 (5.50) 2.12 / 30.74 |

| $A_{old}$ Group | | | | | |
|---|---|---|---|---|---|
| Risk variables | Model 1 OR (SE) $_{95\%}$CI | Model 2 OR (SE) $_{95\%}$CI | Model 3 OR (SE) $_{95\%}$CI | Model 4 OR (SE) $_{95\%}$CI | Model 5 OR (SE) $_{95\%}$CI |
| *Caries experience* | 3.61 (2.06) 1.18 / 11.03 | convergence not achieved | 2.51 (0.92) 1.22 / 5.14 | convergence not achieved | 2.51 (0.92) 1.22 / 5.14 |
| *Related disease* | omitted | | omitted | | omitted |
| *Diet contents* | 5.89 (2.54) 2.52 / 13.72 | | 5.81 (2.32) 2.5 / 12.73 | | 5.81 (2.32) 2.5 / 12.73 |
| *Diet frequency* | 4.16 (1.54) 1.36 / 10.22 | | 3.76 (1.82) 1.44 / 9.72 | | 3.76 (1.82) 1.44 / 9.72 |
| *Plaque amounts* | 3.00 1.60) 1.52 / 10.12 | | 2.34 (0.62) 0.88 / 4.05 | | 1.51 (0.84) 0,90 / 2.13 |
| *Mutans streptococci* | 6.81 (3.56) 2.44 / 18.94 | | 5.29 (2.00) 2.52 / 11.13 | | 5.29 (2.00) 2.52 / 11.13 |
| *Fluoride Program* | 3.16 (1.26) 1.45 / 6.91 | | 0.77 (0.86) 0.08 / 6.82 | | 0.77 (0.86) 0.08 / 6.82 |

(*Continued*)

**Table 4.** (Continued)

| | | | | |
|---|---|---|---|---|
| *Saliva secretion* | 3.46 (0.64) 0.71 / 6.48 | | 2.51 (0.75) 1.00 / 6.08 | 2.51 (0.75) 1.00 / 6.08 |
| *Buffer capacity* | 3.82 (1.14) 0.81 / 4.72 | | 7.28 (1.34) 0.90 / 34.62 | 7.28 (1.34) 0.90 / 34.62 |

assessed and, above all, to the thresholds used to define each category. The allocation biases were perceived for all considered age groups. It is likely that different groups with different caries prevalence and risk variables could lead to different results. The present study findings might serve as a warning to public health dentists, epidemiologists and researchers on the adoption of the Cariogram models with different thresholds may lead to bias and an incorrect interpretation of the subject's real risk. As suggested in the original Cariogram manual, [26, 27] low caries risk is only determined when the "Chance to avoid cavities" is equal to 75% or higher, while a high caries risk is plausible when the 'Chance to avoid cavities' is equal to 25% or lower. No suggestion is given for intermediate values. Nevertheless, in one of the earliest papers published on Cariogram, five risk categories were used (model 5 in the present paper) [27].

From a community or research perspective, an erroneous risk categorization could lead to unnecessary or inadequate preventive interventions [5]. This bias is confirmed by the analysis of the weight of the association between the risk/protective variables and the overall caries risk evaluated through the different five Cariogram models. The background risk factors (*i.e.* related disease, diet content, diet frequency, plaque amounts, oral streptococci, fluoride program, saliva secretion, buffer capacity) had a very different association with the caries risk measured with each model. Hence, the association between the caries risk and the background variables might be miscalculated and misunderstood. For example, the variable plaque amount assumes within the same sample an extremely high weight in some models and a reduced one in others. Also, the role of fluoride program is emblematic; it is not even associated to caries risk in some models (both in children and adults), while strongly associated in some of the adult models, and becomes weaker in others.

On the one hand, the Cariogram assessment at the individual level, allows to graphically evaluate which factors are the heaviest contributors to an individual caries risk, allowing to intervene on them. On the other hand, as previously reported, the graphical advantages of the Cariogram are practically unattainable at a community level, leading to non-effective interventions from a clinical and/or cost / benefit point of view [28, 29].

A major limitation of the present paper is undeniably the small number of participants in the sample $A_{young}$. The small number of participants is unfortunately related to the historical moment, COVID-19 pandemic, in which the study was carried out. As the lockdown was declared by the Italian Government, the study was halted with the hope to resume in September; unfortunately, the conditions related to the pandemic made it impossible, resulting so in a small sample size. Although with this factual limitation, the authors believe that the sample can provide proper support in the paper. In this sample, caries experience showed a quite variable weight in the different models, reaching the highest value in model 3, allowing some generalisation of the results.

Some strengths have to be considered. First of all, this is the first study that compares different models of the Cariogram in different populations. In fact, it is known that the tool does not work as well in all population groups [5, 9]. Furthermore, the fact of having tested different models both in samples of children and adults of different ages gives value to the results of this comparative paper. Finally, having assessed for each model and in each population group the strength of association of each risk factor with the overall caries risk, is a new and little

explored aspect that has allowed to verify how much each factor play a different role in the different models and samples.

## Conclusions

The outcomes of this study might allow to conclude that the caries risk assessment is modified when the Cariogram thresholds vary. The interpretation of caries risk profiles needs to be carefully considered in dental public health, given the risk of misleading the choice of both preventive and management actions, as well as the wrong selection of subjects in specific caries risk categories for research purposes.

## Supporting information

**S1 Checklist. STROBE statement—checklist of items that should be included in reports of** *cross-sectional studies.*
(DOC)

**S1 Appendix.**
(DOC)

**S2 Appendix.**
(DOC)

**S1 Data.**
(XLSX)

## Acknowledgments

The authors want to thank the dental students of the University of Milan that participated in the study.

## Author Contributions

**Conceptualization:** Maria Grazia Cagetti, Guglielmo Campus.

**Data curation:** Giuliana Bontà, Guglielmo Campus.

**Formal analysis:** Guglielmo Campus.

**Methodology:** Maria Grazia Cagetti.

**Supervision:** Maria Grazia Cagetti.

**Writing – original draft:** Maria Grazia Cagetti, Giuliana Bontà, Juan Sebastian Lara, Guglielmo Campus.

**Writing – review & editing:** Maria Grazia Cagetti, Juan Sebastian Lara, Guglielmo Campus.

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
