## [Decision Letter · Decision Letter 0]

2 Nov 2021

PONE-D-21-19761Caries risk assessment using different Cariogram models. A comparative study about concordance in different populations - adults and children.PLOS ONE

Dear Dr. Campus,

Thank you for submitting your manuscript to PLOS ONE. After careful consideration, we feel that it has merit but does not fully meet PLOS ONE’s publication criteria as it currently stands. Therefore, we invite you to submit a revised version of the manuscript that addresses the points raised during the review process.

We look forward to receiving your revised manuscript.

Kind regards,

Denis Bourgeois

Academic Editor

PLOS ONE

Journal Requirements:

3. We noted in your submission details that a portion of your manuscript may have been presented or published elsewhere. [As it was described in the manuscript (Materials and Methods pages 5-6), Four sample groups were considered in the comparisons. Two were obtained from previous studies in children [11, 14], one in adults [15], and the fourth one, which included young adults, was ad hoc enrolled.] Please clarify whether this [conference proceeding or publication] was peer-reviewed and formally published. If this work was previously peer-reviewed and published, in the cover letter please provide the reason that this work does not constitute dual publication and should be included in the current manuscript.

Reviewers' comments:

Reviewer's Responses to Questions

**Comments to the Author**

1. Is the manuscript technically sound, and do the data support the conclusions?

Reviewer #1: No

Reviewer #2: Yes

Reviewer #3: Partly

2. Has the statistical analysis been performed appropriately and rigorously? 

Reviewer #1: No

Reviewer #2: Yes

Reviewer #3: Yes

3. Have the authors made all data underlying the findings in their manuscript fully available?

Reviewer #1: Yes

Reviewer #2: Yes

Reviewer #3: Yes

4. Is the manuscript presented in an intelligible fashion and written in standard English?

Reviewer #1: Yes

Reviewer #2: Yes

Reviewer #3: Yes

5. Review Comments to the Author

Reviewer #1: I have reviewed the paper “Caries risk assessment using different Cariogram models. A comparative study about concordance in different populations - adults and children.”

When assessing caries risk what is expected to forecast is the chance to develop or not the decease. In this sense it would be useful to compare the different models against true caries increment. One would expect that change of cut off points in the Cariogram will end up in different results. Therefore comparing different models between them does not have a clinically meaningful conclusion.

Cariogram weights the different factors using an internal algorithm and produces the outcome. Trying to separate the factors in a regression analysis to identify which correlates with caries risk does not make sense because this has been done by the program. Again, there is no clinically meaningful conclusion for the aim of the study

The introduction does not justify the aim of the study, as literature more relevant to the paper has not been mentioned and the discussion is poor. Regarding the methodology it is not clear why the specific age groups where chosen and if clinical examination was similar to all participants.

Reviewer #2: The reviewed paper is a methodological survey to verify whether there is concordance among several Cariogram risk assessments at different cut-off values, comparing both children and adult populations; and how each risk/protective factor weights on the overall caries risk profile.

For this purpose, a total of three groups' data (two in children and one in adults) were obtained from previous studies, while a fourth, in young adults, was enrolled ad hoc.

The results were significant differences in caries risk stratification using different cut-offs stands out from data analysis; consequently, risk assessments need to be carefully considered due to the risk of misleadingly choosing preventive and research actions.

This is a well-written paper worth publishing in Plos One. However, some points should be addressed:

1. check capitalization -> e.g. keywords or table description table 1.

2. it is not clear whether only the children groups were re-examined and the caries risk recalculated years later or was this re-examined and recalculated for all groups included in the study two years later?

3. the groups are very different in size, especially the sample that was added ad hoc. Even if it is designed for clarity with division young/old children/adults very clearly for the reader, percentages of a sample below 100 are rather strange to read with percentages. Would it not be possible at this point to enlarge this group so that there are not exorbitantly large differences (957, 862, 86, 480)?

4. are there numerous spelling mistakes, e.g. in table 2 the absolute number is missing for caries-free in Aold (n=480) before (36.46%), p. 10 "25% or lower. .", p. 11 "interpretadtion", table description of Table 1, etc.

5. no information on the validation of the questionnaire is given.

6. no information about the training and calibration of the two examiners is given.

7. there is no information about the saliva chair-side test used, e.g. manufacturer, etc.

8. what are the standardized conditions under which the saliva test was used?

9. the correct wording and spelling of the bacteria should be followed.

10. awkward wording "Nevertheless, some of the original Cariogram authors suggested, in one of the earliest papers published on Cariogram, the use of five risk categories) should be reworded.

11) "Plaque amount" is a perfect example of this;" should be reworded.

Reviewer #3: Dear authors,

The study has several significant drawbacks.

The references are outdated.

Only four references are from the last five years. It may help if they find more recent articles as I suggested in a comment in the Introduction.

The conclusion need be more in line with the aim of the research (for abstract and the whole manuscript).

There are recent articles that use different cut-offs to define risk category according to Cariogram. Also, some studies use sensitivity, specificity, positive and negative predictive values according to Youden's index. It is necessary to look for them and expand this part (the Background – part where you mention references 8-10 - in attached documents) with these references, as well as the Discussion part. Taken as a whole, the references from the entire paper are older. Only four references are from the last five years.

The discussion is very brief. The authors compare their results and conclusions with only four references, and one is newer. It seems that they have not considered recent findings of the use of Cariogram.

- What are the strengths of your study?

- Please mention the limitations of your study.

- The authors need to systematically compare their findings with previous studies and provide logical justification for any reported differences.

It may help if they find more recent studies and compare with them as I suggested in comment in the Background.

6. PLOS authors have the option to publish the peer review history of their article (what does this mean?). If published, this will include your full peer review and any attached files.

Reviewer #1: No

Reviewer #2: No

Reviewer #3: No

---

## [Author Response · Author response to Decision Letter 0]

14 Feb 2022

First, we really want to express our gratitude to the Editor and the reviewers for the help provide to us to improve the quality of our paper.

Below all the suggestions form Journal requirements and reviewers (in bold) and our replies (in italics)

Journal Requirements:

Done. 

We included as supplementary file the questionnaire in English.

3. We noted in your submission details that a portion of your manuscript may have been presented or published elsewhere. [As it was described in the manuscript (Materials and Methods pages 5-6), Four sample groups were considered in the comparisons. Two were obtained from previous studies in children [11, 14], one in adults [15], and the fourth one, which included young adults, was ad hoc enrolled.] Please clarify whether this [conference proceeding or publication] was peer-reviewed and formally published. If this work was previously peer-reviewed and published, in the cover letter please provide the reason that this work does not constitute dual publication and should be included in the current manuscript.

Three of the four samples used in this study in order to compare the risk assessment through different Cariogram models have already been the object of study of previous published papers. In this study, the different parameters relating to caries previously collected, were re-evaluated to calculate the risk profiles considering different cutoffs to define each risk categories. As consequence, present findings do not represent a duplicate of what has already been published.

Done

Reviewer #1: 

1. When assessing caries risk what is expected to forecast is the chance to develop or not the disease. In this sense it would be useful to compare the different models against true caries increment. One would expect that change of cut off points in the Cariogram will end up in different results. Therefore, comparing different models between them does not have a clinically meaningful conclusion.

We disagree with the reviewer’s comment. The caries incidence is the only outcome that can establish with certainty if the previous risk assessment was correct. However, in this paper the goal was to verify how, in different samples, the use of different Cariogram models could give the clinician not congruent information. Although all the models tested have been used successfully in the literature, the results of the present evaluation highlight that caution should be used when different cut-offs than those originally proposed by Cariogram’s authors, are used. This does not mean that other models than the original should definitely not be used but that a "calibration" of the model on your each single sample should be performed as suggested in Trottini M, Campus G, Corridore D, Cocco F, Cagetti MG, Vigo MI, Polimeni A, Bossù M. Assessing the Predictive Performance of Probabilistic Caries Risk Assessment Models: The Importance of Calibration. Caries Res. 2020;54(3):258-265.. 

2. Cariogram weights the different factors using an internal algorithm and produces the outcome. Trying to separate the factors in a regression analysis to identify which correlates with caries risk does not make sense because this has been done by the program. Again, there is no clinically meaningful conclusion for the aim of the study.

We disagree with the reviewer’s comment. The internal algorithm weight the different the different factors in a prefixed model, but each population is different and also the weight of the different risk factors in relation to caries increment. We really believe that knowing the weight of the different risk factors on caries increment has a quite important clinical implications especially planning preventive programs in population with skewed caries figures. We will try to explain this in the manuscript. 

3. The introduction does not justify the aim of the study, as literature more relevant to the paper has not been mentioned and the discussion is poor. 

The introduction was modified following the recommendations of the reviewer

4. Regarding the methodology it is not clear why the specific age groups were chosen and if clinical examination was similar to all participants.

We described in the Methodology section this. The clinical examinations were carried out from examiners of the same research group. The research group has an has decades of experience in designing and conducting epidemiological research documented by dozens of scientific publications, some of which focused on methods for calibrating and standardizing examiners for this type of survey. In particular, two authors of the present survey acted as examiners/benchmarks in all surveys: MGC (Ayoung and Aold surveys) and GC (Cyoung and Cold surveys as examiner and Ayoung and Aold as benchmark). We add some clarification in the materials and methods sections. 

Reviewer #2: 

1. Check capitalization -> e.g. keywords or table description table 1.

Capitalization was checked and modified when necessary

2. it is not clear whether only the children groups were re-examined and the caries risk recalculated years later or was this re-examined and recalculated for all groups included in the study two years later?

As suggested by the reviewer this point was clarified. The following sentence was added to Material and Methods (Sample) “Unlike the two groups of adults who are two distinct samples, the two samples of children are actually the same group whose caries risk was evaluated and re-evaluated two years later”

3. the groups are very different in size, especially the sample that was added ad hoc. Even if it is designed for clarity with division young/old children/adults very clearly for the reader, percentages of a sample below 100 are rather strange to read with percentages. Would it not be possible at this point to enlarge this group so that there are not exorbitantly large differences (957, 862, 86, 480)?

The referee's comment hits a very important spot. when we designed the addition of this sample (Ayoung) we had assumed a higher sample. Unfortunately, running the study, the COVID-19 pandemic arrived. The population enrolled in the study were University students in the city of Milan. In April, 2020 the first lockdown was declared, and all university classes went online.

We therefore had to interrupt the study, planning to resume in September; unfortunately, the conditions related to the pandemic forced the academic authorities to keep classes online only.

We are fully aware that this is a major weakness of our study. We decided to add a paragraph about this in the discussion section.

4. are there numerous spelling mistakes, e.g. in table 2 the absolute number is missing for caries-free in Aold (n=480) before (36.46%), p. 10 "25% or lower. .", p. 11 "interpretadtion", table description of Table 1, etc.

We are sorry for this. Spelling mistakes were corrected

5. no information on the validation of the questionnaire is given.

The information about the validation and standardization of the questionnaire was added as a reference.

6. no information about the training and calibration of the two examiners is given.

The information about the examiners’ calibration was added.

7. there is no information about the saliva chair-side test used, e.g. manufacturer, etc.

The information was reported

8. what are the standardized conditions under which the saliva test was used?

“Standardized conditions” was replaced by “following the manufacturer’s instructions”

9. the correct wording and spelling of the bacteria should be followed.

Bacteria names were corrected

10. awkward wording "Nevertheless, some of the original Cariogram authors suggested, in one of the earliest papers published on Cariogram, the use of five risk categories) should be reworded.

The sentence was reformulated as follow: “In one of the earliest papers published on Cariogram, five risk categories were used (model 5 in the present paper)”

11. "Plaque amount" is a perfect example of this;" should be reworded.

The sentence was reformulated, 

Reviewer #3: 

1. The references are outdated. Only four references are from the last five years. It may help if they find more recent articles as I suggested in a comment in the Introduction.

We updated the reference of this, even if the papers about the development of the software has to be cited. 

2. The conclusion need be more in line with the aim of the research (for abstract and the whole manuscript).

We modified the conclusion following the reviewer’s suggestion.

3. There are recent articles that use different cut-offs to define risk category according to Cariogram. Also, some studies use sensitivity, specificity, positive and negative predictive values according to Youden's index. It is necessary to look for them and expand this part (the Background – part where you mention references 8-10 - in attached documents) with these references, as well as the Discussion part. Taken as a whole, the references from the entire paper are older. Only four references are from the last five years.

As described above we updated the references and modified both the introduction and the discussion section following the reviewer’s suggestions.

4. The discussion is very brief. The authors compare their results and conclusions with only four references, and one is newer. It seems that they have not considered recent findings of the use of Cariogram. What are the strengths of your study?

A deep re-writing of the discussion section was performed. We also include the strengths of the study.

5. Please mention the limitations of your study.

We mentioned the limitation of our study in the discussion section. 

6. The authors need to systematically compare their findings with previous studies and provide logical justification for any reported differences. It may help if they find more recent studies and compare with them as I suggested in comment in the Background.

We followed the reviewer’s suggestions and we deeply modified both the introduction and the discussion sections including the limitations of the study and comparing the findings with previous publications.

---

## [Decision Letter · Decision Letter 1]

21 Feb 2022

Caries risk assessment using different Cariogram models. A comparative study about concordance in different populations - adults and children.

PONE-D-21-19761R1

Dear Dr. Campus,

We’re pleased to inform you that your manuscript has been judged scientifically suitable for publication and will be formally accepted for publication once it meets all outstanding technical requirements.

Kind regards,

Denis Bourgeois

Academic Editor

PLOS ONE

Additional Editor Comments (optional):

Reviewers' comments:

Reviewer's Responses to Questions

**Comments to the Author**

1. If the authors have adequately addressed your comments raised in a previous round of review and you feel that this manuscript is now acceptable for publication, you may indicate that here to bypass the “Comments to the Author” section, enter your conflict of interest statement in the “Confidential to Editor” section, and submit your "Accept" recommendation.

Reviewer #2: All comments have been addressed

Reviewer #3: All comments have been addressed

2. Is the manuscript technically sound, and do the data support the conclusions?

Reviewer #2: Yes

Reviewer #3: Yes

3. Has the statistical analysis been performed appropriately and rigorously? 

Reviewer #2: Yes

Reviewer #3: Yes

4. Have the authors made all data underlying the findings in their manuscript fully available?

Reviewer #2: Yes

Reviewer #3: Yes

5. Is the manuscript presented in an intelligible fashion and written in standard English?

Reviewer #2: Yes

Reviewer #3: Yes

6. Review Comments to the Author

Reviewer #2: (No Response)

Reviewer #3: The authors corrected the article according to the reviewer's suggestions.

The paper can be accepted for publication.

7. PLOS authors have the option to publish the peer review history of their article (what does this mean?). If published, this will include your full peer review and any attached files.

Reviewer #2: No

Reviewer #3: No

---

## [Editor Report · Acceptance letter]

16 Jun 2022

PONE-D-21-19761R1 

Caries risk assessment using different Cariogram models. A comparative study about concordance in different populations - adults and children.

Dear Dr. Campus:

I'm pleased to inform you that your manuscript has been deemed suitable for publication in PLOS ONE. Congratulations! Your manuscript is now with our production department. 

Kind regards, 

on behalf of

Professor Denis Bourgeois 

Academic Editor

PLOS ONE